# Hexacyano Ferrate (III) Reduction by Electron Transfer Induced by Plasmonic Catalysis on Gold Nanoparticles

**DOI:** 10.3390/ma12183012

**Published:** 2019-09-17

**Authors:** Iyad Sarhid, Isabelle Lampre, Diana Dragoe, Patricia Beaunier, Bruno Palpant, Hynd Remita

**Affiliations:** 1Laboratoire de Chimie Physique, Université Paris-Sud, UMR 8000 CNRS, Université Paris-Saclay, 91405 Orsay, France; iyad.sarhid@u-psud.fr; 2Institut de Chimie Moléculaire et des Matériaux, Université Paris-Sud, UMR 8182 CNRS, Université Paris-Saclay, 91405 Orsay, France; diana.dragoe@u-psud.fr; 3Sorbonne Université, CNRS, Laboratoire de Réactivité de Surface, UMR 7197, F-75005 Paris CEDEX 05, France; patricia.beaunier@sorbonne-universite.fr; 4Laboratoire de Photonique Quantique et Moléculaire, UMR 8537 CentraleSupélec/Ecole Normale Supérieure Paris-Saclay/CNRS, Université Paris Saclay, 91190 Gif-sur-Yvette, France; bruno.palpant@centralesupelec.fr; 5CNRS, Laboratoire de Chimie Physique, UMR 8000 Université Paris-Sud, Université Paris-Saclay, 91405 Orsay, France

**Keywords:** catalysis, ferricyanide (III) reduction, plasmon resonance, gold nanoparticles, hot electrons, electron transfer, size effect

## Abstract

Redox reactions are of great importance in environmental catalysis. Gold nanoparticles (Au-NPs) have attracted much attention because of their catalytic activity and their localized surface plasmon resonance (LSPR). In the present study, we investigated, in detail, the reduction of ferricyanide (III) ion into a ferrocyanide (II) ion catalyzed by spherical gold nanoparticles of two different sizes, 15 nm and 30 nm, and excited at their LSPR band. Experiments were conducted in the presence (or absence) of sodium thiosulfate. This catalysis is enhanced in the presence of Au- NPs under visible light excitation. This reduction also takes place even without sodium thiosulfate. Our results demonstrate the implication of hot electrons in this reduction.

## 1. Introduction

Redox reactions are of great importance in environmental catalysis, such as, for example, the reduction of nitrogen oxides (deNOx), CO_2_, or heavy metals, and the generation of hydrogen from water or oxidation of CO or organic pollutants [1]. Platinum, palladium, and rhodium are the most used metal catalysts in hydrogenation reactions or in catalytic convertors for deNOx and CO reduction [2]. Catalysis by gold has been a domain of increased interest since the discovery by Haruta [3] that small gold nanoparticles (Au-NPs) can catalyze CO reduction at room temperature [4,5,6,7,8,9].

Au-NPs have also attracted much attention due to their localized surface plasmon resonance (LSPR), i.e., the oscillations of metal free electrons driven by the electric field of the incident light. There are many applications utilizing the SPR of gold nanoparticles, including subwavelength electromagnetic energy transport, chemical and biological sensors [1,6,10], surface enhanced Raman scattering (SERS) [11,12], plasmonic photocatalysis [13,14,15], and photothermal cancer therapy [16,17,18], where the plasmonic energy is converted locally into heat that raises the surrounding medium temperature, leading to the killing of cancer cells. In thermal therapy, hot electrons induced by surface plasmon excitation lead (after reaction with oxygen or water) to the formation of reactive oxygen species (ROS) (HO, O_2_^−^ and singlet oxygen), which are also responsible for cancer cell killing [19]. 

Metal nanoparticles, such as gold and platinum NPs, are efficient catalysts in different reactions with environment applications. These catalytic reactions often involve electron transfers [20,21]. The reduction of ferrocyanate (III) has been taken as a model reaction to study catalysis by metal NPs [20,21,22]. 

Mulvaney et al. [22] studied redox catalysis by colloidal gold for the reduction of ferricyanide (III) ions by borohydride ions in alkaline aqueous solution. They found that Au-NPs dramatically increase the reaction rate and that Au-NPs also change the mechanism. The reaction is zero order with respect to the hexacyanoferrate (III) concentration for the non-catalyzed reaction and first order for the catalyzed reaction. In this reaction, the Au-NPs act as a reservoir for the electrons and become cathodically polarized. In the first step, borohydride injects electrons onto Au-NPs, and then ferricyanide ions diffuse to the nanoparticle surface and are reduced by excess surface electrons. These nanoparticles serve as a relay for electrons.

El Sayed‘s group reported on catalysis by metal nanoparticles for the reduction of ferricyanide ion by thiosulfate to ferrocyanide Equation (1) [20,21,23,24,25]: (1)$$2\mathrm{Fe}{\left(\mathrm{CN}\right)}_{6}^{3-}+\;2S_{2}O_{3}^{2-}\to \;2\mathrm{Fe}{\left(\mathrm{CN}\right)}_{6}^{4-}+S_{4}O_{6}^{2-}.$$

This reaction was used to compare the catalytic activity of different nanoparticles. The size and shape dependent catalytic activity of platinum nanoparticles for this reaction was demonstrated [23,24,25]. The study conducted on tetrahedral, cubic, and spherical Pt nanoparticles demonstrated that the kinetic parameters correlate with the fraction of surface atoms located on the corners and edges in each size and shape [23]. It was also demonstrated that Au-NPs are very efficient catalysts for the reduction of ferricyanide (III) ion to ferrocyanide. El-Sayed et al. reported that catalysis of this reaction by gold nanoparticles was enhanced when the latter were irradiated at the plasmon wavelength [21]. The value of the activation energy of this electron transfer reaction for two different shapes of Au-NPs (nanospheres and nanocages) was measured. This activation energy was found to be similar when using a thermostatic water bath or by using a plasmonic photothermal reactor. This result supported the conclusion that the role of the surface plasmon field on the electron dynamics during the reaction was negligible and that the main effect was due to a temperature increase after plasmon excitation [7,14,15].

Recently, the LSPR effect of plasmonic nanostructures was successfully applied to photocatalysis under visible light irradiation and proved to be very promising [14,15,26]. 

In the present study, we investigated in details the reduction of ferricyanide (III) ion into ferrocyanide (II) ion catalyzed by spherical Au-NPs and excited at their LSPR band. Experiments were conducted in the presence and in the absence of sodium thiosulfate. This catalysis is enhanced in the presence of Au-NPs with light excitation. We show for the first time that this reduction takes place even without sodium thiosulfate, thereby demonstrating the implication of hot electrons in this reduction. 

## 2. Materials and Methods 

Tetrachloroauric acid (HAuCl_4_ 3H_2_O), trisodium citrate trihydrate, potassium hexacyanoferrate (III) (HC-FeIII), sodium thiosulfate (ST), and para-nitrothiophenol (pNTP) of high purity were purchased from Aldrich and used as received. All the experiments were carried out with Milli-Q water with resistivity higher than 18 MΏ cm.

Spherical Au-NPs were synthesized by the Türkevich method [27,28,29,30,31]. An aqueous solution of HAuCl_4_ was heated to the boiling point in an Erlenmeyer flask, and then, a sodium citrate aqueous solution was added with vigorous magnetic stirring. This solution was heated for about 10 min, and the red solution obtained was left to cool. Two well-defined sizes of Au-NPs (Au-NPs@citrate) were obtained by changing the experimental conditions. Au-NPs with mean diameters of 15 and 30 nm, respectively, were obtained with a concentration ratio of citrate to gold equal to 6.8 and 1.4, respectively, and an initial concentration of gold ions, [Au^III^], of 0.25 mM and 1 mM, respectively. 15 nm (30 nm) Au-NP@citrate colloidal solutions were produced by adding 1 mL (0.8 mL) of an aqueous solution of trisodium citrate (34 mM, 1 wt %) to 20 mL of a stirred boiling aqueous solution of HAuCl_4_ of concentration 0.25 mM (1 mM) [30]. The 30 nm-Au-NPs@citrate solutions were then diluted by a factor of four to get the same concentration of Au atoms as in the 15 nm-Au-NPs@citrate solutions.

To change the stabilizing agent, aqueous solutions of pNTP (6 × 10^−4^ M) were prepared, and the pH was adjusted to 11 by the addition of 0.1 M NaOH solution in order to form the phenolate anion Then, 750 µL of an alkaline pNTP solution were added to 2 mL of AuNP solutions contained in a 1 cm square cuvette. Following that, the solutions were left under continuous stirring in the dark for 40 min to ensure the adsorption of pNTP and the replacement of citrate on the surface of the Au-NPs. The functionalized nanoparticles were then centrifuged and further dispersed for catalytic experiments.

For the catalytic reactions under irradiation, sodium thiosulfate and hexacyano-ferrate (III) were added to a solution containing Au-NPs. The final concentrations for each reagent were: sodium thiosulfate: 5 × 10^−4^ M, hexacyano-ferrate (III): 5 × 10^−4^ M, Au-NPs: 0.25 × 10^−3^ M (concentration in Au atoms). Next, 2 mL of the mixture were introduced in a quartz cell with an optical path of 1 cm (and containing a small magnetic rod). The cell was properly sealed, bubbled with N_2_ for 20 min to get rid of O_2_, and then irradiated under stirring either with LEDs at λ = 520 nm (home reactor, please find it in Appendix A) or with an Oriel 300 W Xenon lamp (Heraeus, Hanau, Germany) equipped with an infrared water filter and an optical cutoff filter (longpass filter GG450, Edmund optics, Lyon, France, λ > 450 nm) to avoid the direct excitation of the ferrocyanide (III) ion, which absorbs in the range of 350–400 nm (Appendix A). The reactions were conducted at room temperature (22 °C). A thermocouple was used to monitor the temperature during the reaction. Due to the absorption of light by all the Au-NPs in the solution, the temperature of the latter was raised to 26 °C maximum. 

The UV-visible absorption spectra were recorded using a HP8453 spectrophotometer (Agilent technologies, Les Ulis, France) and were used to follow the reaction kinetics.

The size and shape of the Au-NPs before and after catalytic reactions were characterized by Transmission Electron Microscopy (TEM) using a JEOL JEM 100CX microscope (JEOL, Tokyo, Japan) operating at an accelerating voltage of 100 kV. A few drops of the samples (which were first sonicated to disperse the precipitated aggregates when necessary) were deposited on carbon coated copper grids and dried under an N_2_ flow. TEM images were taken from different sections of the TEM grid to verify the size and shape distributions of the nanoparticles.

XPS analysis was performed using a K Alpha instrument from Thermo Fisher Scientific, East Grinstead, UK, with a monochromatic Al_Kα_ X-ray source (1486.7 eV) and a hemispherical analyzer, at a take-off angle of 0°. Films of the samples were drop-cast on thoroughly cleaned silica plates. The wide scan spectra were acquired at a pass energy of 200 eV and an energy step of 1 eV, while the narrow scan spectra were recorded at 50 eV pass energy and 0.1 eV energy step. Charge correction was accomplished by means of a dual beam source. The binding energy scale was checked against neutral C1s, set at 284.8 eV. Binding energies are given with a precision of ± 0.2 eV. The spectra obtained were treated by means of the Avantage software, version 5.967 provided by Thermo Fisher. A Shirley background subtraction was used, and the peak areas were normalized using the Scofield sensitivity factors. Mixed Gaussian-Lorentzian (GL) functions (30% L) were used as the line shapes in the spectral decomposition, with asymmetry added to the metallic Au core-level spectrum. 

## 3. Results

We have investigated the catalytic properties of Au-NPs under plasmon excitation on the reduction of HC-FeIII in the presence or in the absence of the reducing ST. Figure 1 shows the UV-visible spectra and TEM images of the synthesized and used Au-NPs@citrate with two different main diameters of 15 and 30 nm. The maximum of the LSPR band is located at 515 and 525 nm for the solutions containing Au-NPs, with a mean size of 15 and 30 nm, respectively. 

### 3.1. Reduction of Hexacyanoferrate in the Presence of Sodium Thiosulfate

The studied reaction is the reduction of ferricyanide or hexacyanoferrate, HC-FeIII, by thiosulfate Equation (2):(2)$$2\;\mathrm{Fe}{\left(\mathrm{CN}\right)}_{6\;\left(\mathrm{aq}\right)}^{3-}+2{\;S}_{2}O_{3\;\left(\mathrm{aq}\right)}^{2-}\to \;2\;\mathrm{Fe}{\left(\mathrm{CN}\right)}_{6\;\left(\mathrm{aq}\right)}^{4-}+S_{4}O_{6\;\left(\mathrm{aq}\right)}^{2-}.$$

The redox potential of the couples involved in this reaction are E^0^(Fe(CN)_6_^3−^/Fe(CN)_6_^4−^) = +0.36 V and E^0^(S_4_O_6_^2−^/S_2_O_3_^2−^) = +0.08 V versus the normal hydrogen electrode at 25 °C [32]. Although thermodynamically favorable, this reaction needs to be catalyzed. Solutions containing HC-FeIII and ST (without Au-NPs) are stable at room temperature, even under irradiation, while in the presence of Au-NPs and in the absence of irradiation, the reaction takes place, as shown by the disappearance of the absorption bands at 310 and 420 nm associated to HC-FeIII (Figure 2). The rate of the reaction depends on the size of the Au-NPs (Figure 3). Catalysis is a surface phenomenon. Higher reduction kinetics are obtained with smaller gold nanoparticles, which exhibit a much larger surface area and more sites of low coordination (edges and corners), which are important in catalytic steps. Considering the average size of the nanoparticles (15 and 30 nm), and assuming that they are quasi-spherical, their specific surface areas are 14.5 m^2^ g^−1^ and 7.3 m^2^ g^−1^. In agreement with Mulvaney’s study [22], the catalyzed reaction is first order with respect to HC-FeIII, as evidenced by the plot of the logarithm of the absorbance of HC-FeIII versus time (Figure 3, bottom), and the deduced values of the rate constant are 0.0534 and 0.0244 min^−1^ for the 15 nm and 30 nm AuNPs@citrate, respectively.

#### 3.1.1. Irradiation Using LEDs at λ = 520 nm

Figure 2 presents the spectral evolution of solutions containing HC-FeIII, ST and Au-NPs@citrate under LEDs irradiation at λ = 520 nm, close to the LSPR of Au-NPs. For both sizes of Au-NPs@citrate, the bands at 310 and 430 nm, due to HC-FeIII, decrease in intensity with irradiation time and disappear after about 45 and 60 min of irradiation for 15 nm and 30 nm Au-NPs@citrate, respectively. The LSPR band of the spherical Au-NPs (with a maximum around 515 nm or 525 nm for the solutions containing Au-NPs of 15 and 30 nm, respectively) remains quite stable during the whole irradiation time, indicating no drastic changes in Au-NP morphology.

The kinetics of the reduction reaction of HC-FeIII by ST are followed by a decrease in the absorbance at 420 nm (Figure 3). The reduction is faster for the smaller 15 nm Au-NPs@citrate compared to 30 nm Au-NPs@citrate, and for both sizes, the reduction is accelerated by the irradiation. Under irradiation, the reaction kinetics follow the same order as without irradiation (i.e., an apparent first order law with respect to HC-FeIII). Taking into account only the data for the first 20–30 min, the initial rate constant values are determined to be 0.0669 and 0.0475 min^−1^ for 15 nm and 30 nm AuNPs@citrate, respectively. Indeed, the small increase in temperature measured during the reaction time (≤4 °C) might result in a slight increase in the reaction rate during the reaction time, leading to an overestimate of the global rate constant, as suggested by the linear fit in Figure 3 (bottom). Moreover, the small changes in the LSPR band of the AuNPs@citrate that were observed during the reaction (Figure 3 (top)) also introduce uncertainties into the determination of the absorbencies due to HC-FeIII, which may affect the values of the rate constant.

The excitation of the LSPR of the Au-NPs favors the catalytic properties of the Au-NPs. This plasmonic effect could be related to different mechanisms: Electron density exchange between the excited gold nanoparticles and nearby reactants;Effect of the plasmonic near field enhancement on the electron transfer process between HC-FeIII and ST;Temperature increase due to a rapid conversion of the excited surface plasmon resonance into heat that would eventually accelerate the chemical reaction.

Notably, the temperature increased in the solution during the reaction under irradiation and stirring was only 3–4 degrees, as measured by a thermo-couple immersed inside the reaction cuvette. Under continuous irradiation, each NP contributed to the global warming of the solution they were surrounded by. As the irradiation was continuous, and considering the NPs’ size, light power, stirring, and irradiation duration, the thermal gradient in the vicinity of each NP was negligible, so the local overheating effect can be disregarded [33]. Consequently, the increase of temperature due to plasmon excitation is not enough to favor the reduction power of citrate.

#### 3.1.2. Irradiation Using a Xe Lamp Equipped with a 450 nm Optical Cutoff Filter

Similar experiments were performed using a Xe lamp with a 450 nm optical cutoff filter instead of using the LEDs at 520 nm. Under these conditions, in addition to the disappearance of the absorption bands at 310 and 420 nm due to HC-FeIII, changes in the LSPR band of the Au-NPs@citrate were observed (Figure 4). For 15 nm Au-NPs@citrate, the LPSR band around 530 nm fully disappears simultaneously (as with the bands of HC-FeIII (Figure 4a)), while for the 30 nm Au-NPs@citrate, the LSPR band around 530 nm decreased partially, and a new broad band with a maximum of around 750 nm appears (Figure 4b). These results indicate that during the reduction reaction upon the Xe lamp’s irradiation, the morphology of the Au-NPs@citrate changes dramatically, what is confirmed by TEM images (Figure 5 and Figure 6).

Figure 5 and Figure 6 show TEM images of the Au-NPs present at different times during the reduction reaction under Xe lamp irradiation. The TEM images show that the Au-NPs reshape with irradiation time. They become more elongated and assemble in necklaces. This reshaping might be due to heating of the Au-NPs by plasmon excitation and to reactions (which are probably catalyzed by the plasmon) at the surface of the nanoparticles, leading to deligandation of the citrate ions and to chemical complexation of the Au with cyanide. These phenomena depend on the initial size of the Au-NPs. Indeed, in the case of the initial 15 nm Au-NPs@citrate, the UV-visible spectra do not exhibit a new plasmon band, which could correspond to elongated nanoparticles or their assembly, in contrast to what is observed for the 30 nm Au-NPs@citrate. This suggests that for the initial smallest 15 nm Au-NPs@citrate, assembly and precipitation of the elongated Au-NPs takes place rapidly (in seconds or minutes, at the scale of the record of the spectra), while for the initial 30 nm Au-NPs@citrate, smaller aggregates are formed and remain in the suspension. 

### 3.2. Reduction of Hexacyanoferrate in the Absence of Sodium Thiosulfate

#### 3.2.1. Irradiation Using LEDs at λ = 520 nm

Irradiation of solutions containing HC-FeIII and Au-NPs@citrate, but without ST, leads to a decrease in the absorption bands at 310 and 420 nm associated with HC-FeIII, as well as a decrease in the LSPR of the Au-NPs around 520 nm (Figure 7). However, this evolution is very slow compared to the changes observed in the presence of ST under Xe lamp irradiation: While the total reduction of HC-FeIII occurs in less than 1 hour with ST (Figure 3), hardly anything happens for 2 hours without ST (Figure 8). However, the decrease in absorbance is faster for the 15 nm Au-NPs@citrate than for 30 nm Au-Nps@citrate (Figure 8). In the dark, a tiny decrease in absorbance is observed, indicating the absence of a reaction and the stability of the solution without irradiation (Appendix A). 

The absorbance changes observed under irradiation indicate that reactions that are related to the reduction of HC-FeIII occur. We show here for the first time that this reduction takes place even without a reducing agent, such as thiosulfate or borohydrate. In the presence of TS, the reaction is faster, and TS plays the role of an electron relay. Without TS, the reduction reaction is likely due to hot electrons produced by plasmon excitation.

#### 3.2.2. Irradiation Using a Xe Lamp Equipped with a 450 nm Optical Cutoff Filter

Figure 9 presents the spectral evolution of a deaerated solution containing HC-FeIII and 15 nm Au-NPs@citrate under irradiation by a Xe lamp with a longpass filter at 450 nm. As in the case of the LED irradiation (Figure 8), a decrease in the absorbance of the bands at 310, 420, and 520 nm is observed, but this decrease is much faster since, after 3 hours, the bands are almost vanished. However, a comparison between Figure 4a and Figure 9 indicates that the presence of ST accelerates the spectral evolution recorded under Xe lamp irradiation and, consequently, accelerates the reaction. Even without ST, however, reshaping and precipitation due to the aggregation of Au-NPs@citrate continue to proceed. 

#### 3.2.3. Effect of the Stabilizing Agent of the Au-NPs

In light of the results presented above, we repeated the experiments using Au-NPs stabilized by a non-reducing agent, so we replaced the citrate by para-nitrothiophenol (pNTP), since thiol based compounds are well-known and used as stabilizing agents for gold nanoparticles, due to the strong affinity between the sulfur and gold atoms [34]. However, one must keep in mind that during the reaction, the stabilizing agent (pNTP) could also transform, and lead to the formation of *para*-aminothiophenol (pATP), but without a significant change in the stabilization of the Au-NPs [35].

Figure 10 shows the temporal evolution of the UV-visible spectrum of a deaerated solution containing 15 nm Au-NPs@pNTP and HC-FeIII under LED irradiation at 520 nm. In this case, while the band at 420 nm due to HC-FeIII drops, no change is observed in the LSPR band of the Au-NPs. This result indicates that the 15 nm Au-NPs@pNTPs are stable during the reaction under irradiation, which is confirmed by TEM images (Figure 11). It is also worth noting that the disappearance of the absorption band due to Hc-FeIII is quite rapid, nearly as fast as that in the presence of ST (Figure 2c). Citrate and para-nitrothiophenol are weak reducing agents. In the dark, the reaction does not proceed without thiosulfate for both AuNPs@citrate and AuNPs@pNTP. This indicates that the reduction is not due to para-nitrothiophenol or citrate. Furthermore, the increase in temperature due to plasmon excitation is not enough to favor the reduction power of citrate or that of para-nitrothiophenol [33]. These experiments prove that, even without a reducing agent, the reduction of HC-FeIII occurs via the plasmon excitation of Au-NPs, and, consequently, hot electrons are involved.

## 4. Discussion

The previous results show the disappearance of HC-FeIII catalyzed by Au-NPs under irradiation in the presence, or absence, of ST. This disappearance is attributed to the reduction of HC-FeIII. To confirm the reaction and to attain some insights into the role played by Au-NPs, XPS analyses were carried out to determine the oxidation states of iron and gold. 

The XPS wide-scan spectra of 30 nm Au-NPs@citrate after the reaction under LED irradiation at 520 nm for deaerated solutions containing HC-FeIII with ST or without ST (Appendix A) show the presence of gold, iron, carbon, nitrogen, oxygen, potassium, and sodium. Oxygen and carbon are due to citrate ions on the surface of the Au-NPs. Sodium can be related to sodium citrate but also to sodium thiosulfate when present, and potassium and nitrogen come from potassium hexacyanoferrate. Figure 12 presents the XPS spectra recorded for the Au4f, Fe2p, and N1s regions. 

The main Au4f signals appear as a doublet separated by 3.7 eV, with Au4f_7/2_ and Au4f_5/2_ peaks at 83.6 and 87.3 eV, respectively [36], which are characteristic values of binding energies for metallic Au atoms (Figure 12a,b). In the case of the sample without ST, a second doublet is identified at 85.1eV and 89.0 eV (Figure 12b), which can be attributed to the Au–CN bounds [37,38], rather than to Au-citrate [38,39,40]. This assumption is supported by the presence of a contribution at 398.7 eV in the N1s spectrum (Figure 12f), which has an area equivalent to that of Au^+^ in Au4f [37]. This component at 85.1 eV is also present in the Au4f core-level spectrum of the sample with ST (Figure 12a) but to a lesser extent, proving that Au is more protected from oxidation when ST is present. Investigations of Fe2p core-level spectra further corroborate the effect of ST. A complete reduction of Fe^III^ into Fe^II^ is observed in the presence of thiosulfate (Figure 12c), while in the absence of thiosulfate, only 84% of Fe^III^ has been reduced (Figure 12d). In Figure 12c, Fe^II^ is easily identified by the presence of a single doublet located at 708.5 (Fe2p_3/2_) and 721.5 eV (Fe2p_1/2_), typical values for K_4_[Fe(CN)_6_] [41]. This result confirms the hypothesis that, in the presence of ST, at the end of the reaction (after LED irradiation for 60 min), there is a complete reduction of Fe^III^ into Fe^II^ in the present experimental conditions. The presence of a second doublet at 709.8 and 723.7 eV in the spectrum of the sample without ST (Figure 12d) is indicative of Fe^III^ [42,43]. The N1s core-level spectrum in Figure 12e was decomposed into three contributions, one corresponding to nitrogene in FeII-CN bridges, at 397.7 eV, a second contribution at 398.6 eV, which can be attributed to AuCN [29], and a small contribution at 400.4 eV, corresponding to charged nitrogen. In the case of the Au-NPs@citrate without sodium thiosulfate (Figure 12f), an additional component corresponding to Fe^III^-CN is considered. The separation between the contributions Fe^II^-CN and Fe^III^-CN to N1s is considered to be 0.4 eV [https://srdata.nist.gov/xps], with a ratio in agreement with that of Fe^III^/Fe^II^ obtained from the Fe2p spectrum. 

The components attributed to AuCN (in the sample with or without ST) show that the reaction takes place at the surface of Au-NPs@citrate, leading to an oxidation of Au atoms and ligation of Au-NPs@citrate to CN^−^. This oxidative ligandation by CN^−^ might account for the small changes observed in the LSPR of Au-NPs@citrate in the presence of ST and the decrease in the LSPR band in the absence of ST (Appendix A), oxidation being more effective in that case. Indeed, it is well-known that cyanides are used to extract gold from mines via oxidation and complexation [44], as CN^−^ has a great ability to complex with Au and form a stable complex Au(CN)_2_^−^.

In TEM images, the small clusters of AuNPs indicate this partial dissolution. Similar dissolution by cyanide ions has already been invoked in the case of the size reduction and reshaping of the Pt NPs used as catalysts for an electron transfer reaction between hexacyanoferrate and thiosulfate [21,22]. 

To summarize, in the presence of ST, Au is much less oxidized than in the absence of ST, because, in the former case, the Au-NPs@citrates work mainly as catalysts in the electron transfer from thiosulfate to Fe^III^. When ST is not present, the reduction of Fe^III^ occurs, as shown by UV-visible spectra and XPS analysis. This reduction takes place by hot electrons induced by plasmon excitation of Au-NPs. The electrons are mainly transferred from Au NPs to Fe^III^, leading to Au^I^, which will probably be ligated to the CN^−^ present in the solution. 

## 5. Conclusions

Plasmonic catalysis enables us to achieve reactions using solar light with less energy and time consumption. The reduction kinetics of the ferricyanide (III) ion into ferrocyanide (II) by Au-NPs is enhanced under visible light excitation. The reduction kinetics increase with a decreasing nanoparticle size, because of the higher surface areas and greater number of sites of low coordination.

We report here for the first time that this reduction takes place even without sodium thiosulfate and is due to transfer of hot electrons induced by plasmon excitation. The Au-NPs stabilized by citrate reshape during the reaction. However, the shape of the Au-NPs remains stable when they are stabilized by stronger ligands, such as para-nitrothiophenol, and the Au-NPs could then be reused in different catalytic cycles.

This study sheds some light on the possible direct applications of catalysis assisted by plasmon using gold nanoparticles excited by visible light. The results demonstrate the effectiveness of using colloidal spherical gold nanoparticles to catalyze redox reactions under visible irradiation. Further work will focus on anisotropic Au nanoparticles (such as Au nanorods or Au nanostars), which absorb in a larger spectral range compared to spherical Au NPs and could be used to better harvest solar light for catalytic applications [44]. 

## Figures and Tables

**Figure 1 materials-12-03012-f001:**
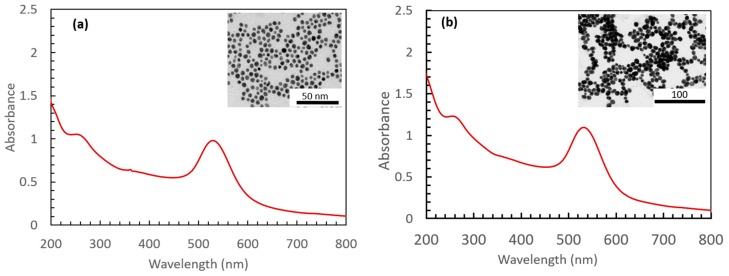
UV-Visible spectra and inset TEM images of the spherical gold nanoparticles synthesized by citrate reduction: (**a**) 15 nm-gold nanoparticles (AuNPs)@citrate and (**b**) 30 nm-AuNPs@citrate. Optical path length: 1 cm.

**Figure 2 materials-12-03012-f002:**
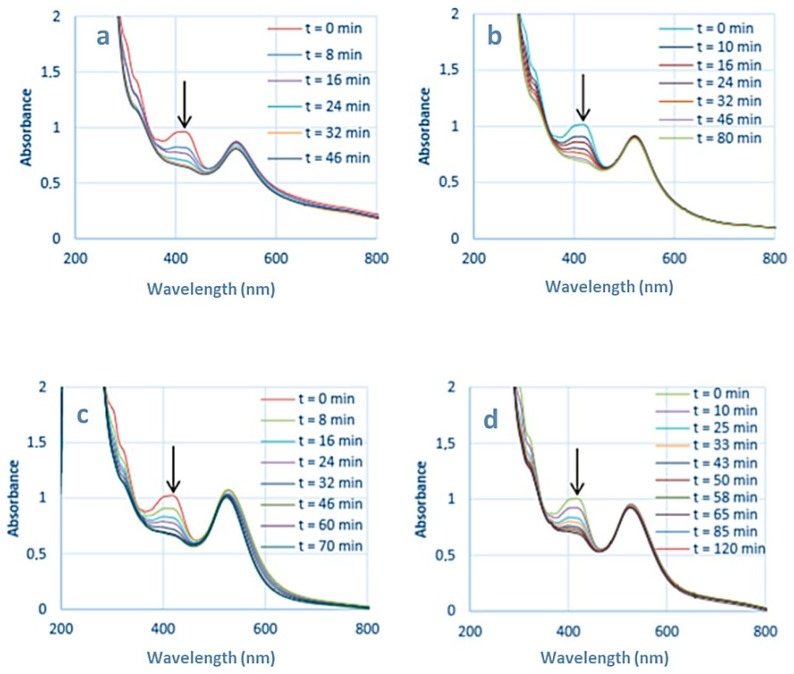
Temporal evolution of the UV-visible spectra of deaerated aqueous solutions containing 5 × 10^−4^ M sodium thiosulfate and 5 × 10^−4^ M potassium hexacyanoferrate: (**a**) in the presence of 15 nm Au-NPs@citrate under irradiation at λ = 520 nm; (**b**) in the presence of 15 nm Au-NPs@citrate in the dark; (**c**) in the presence of 30 nm Au-NPs@citrate under irradiation at λ = 520 nm, and (**d**) in the presence of 30 nm Au-NPs@citrate in the dark. Optical path length: 1 cm.

**Figure 3 materials-12-03012-f003:**
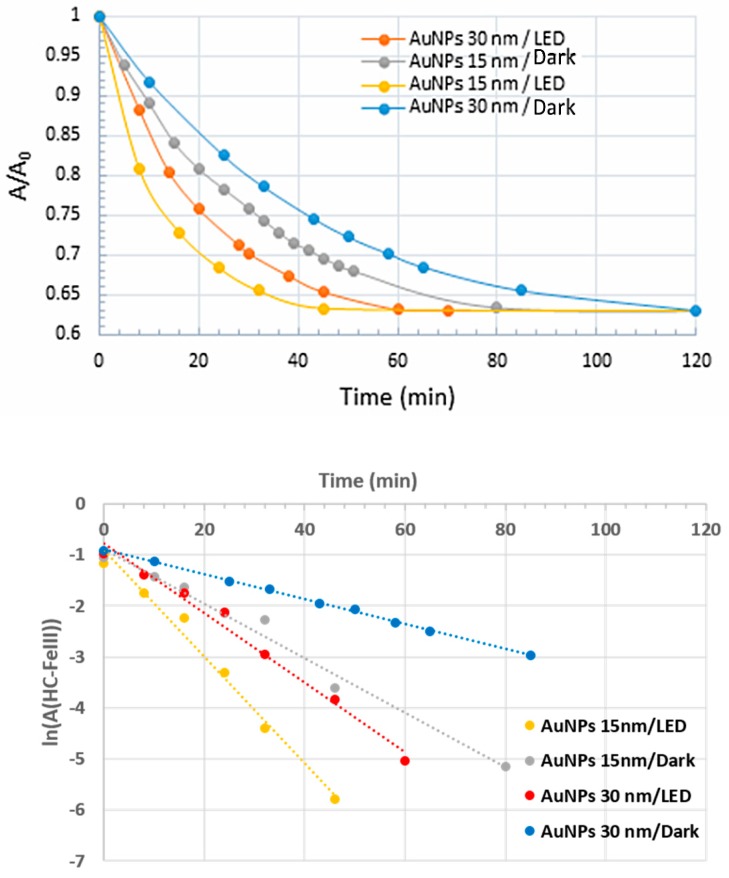
Kinetic traces of the normalized absorbance at 420 nm (**top**) and logarithm of the absorbance of HC-FeIII at 420 nm (**bottom**) for the reduction of HC-FeIII in a deaerated aqueous solutions containing 5 × 10^−4^ M sodium thiosulfate and 5 × 10^−4^ M potassium hexacyanoferrate in the presence of 15 nm or 30 nm Au-NPs@citrate in the dark or under LEDs (irradiation at 520 nm).

**Figure 4 materials-12-03012-f004:**
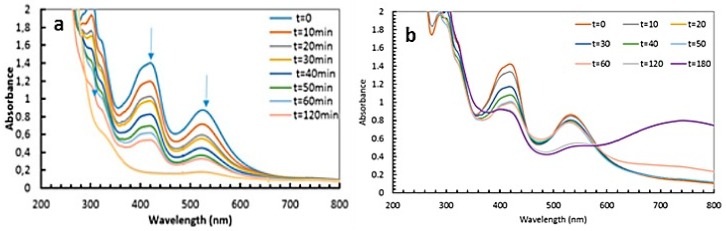
Temporal evolution of the UV-visible spectra of deaerated aqueous solutions containing 5 × 10^−4^ M sodium thiosulfate and 5 × 10^−4^ M potassium hexacyanoferrate under Xe lamp irradiation (equipped with an optical cutoff filter at 450 nm) (**a**) in the presence of 15 nm Au-NPs@citrate and (**b**) 30 nm Au-NPs@citrate. Optical path length: 1 cm.

**Figure 5 materials-12-03012-f005:**
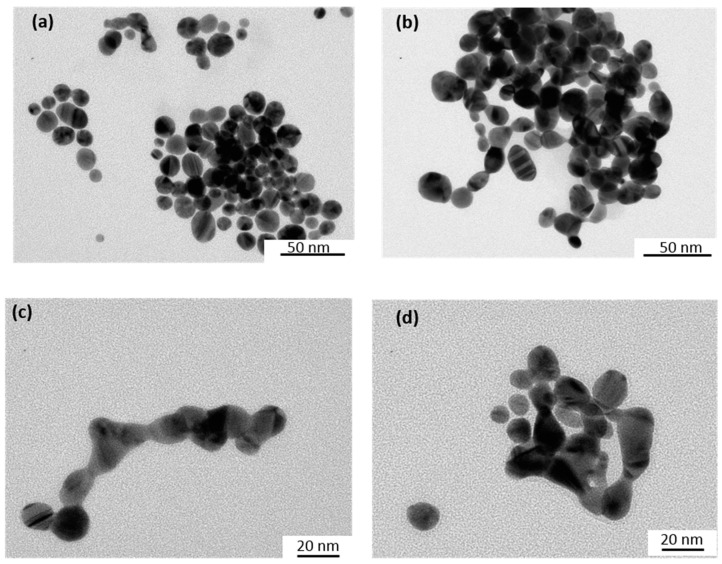
TEM images of the initial 15 nm Au-NPs@citrate taken at different times during the reduction reaction of hexacyanoferrate (III) by sodium thiosulfate under irradiation with a Xe lamp equipped with a cutoff filter at 450 nm: (**a**) before irradiation; after (**b**) 30 min; (**c**) 120 min; and (**d**) 180 min of irradiation.

**Figure 6 materials-12-03012-f006:**
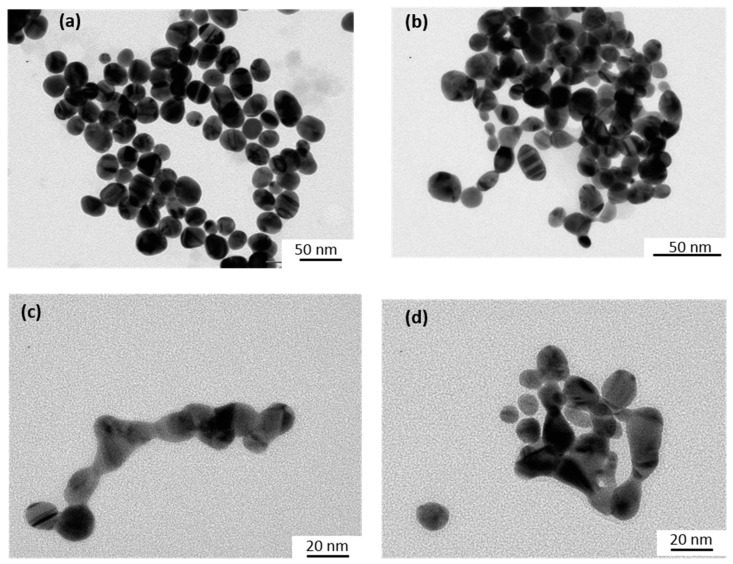
TEM images of initial 30 nm Au-NPs@citrate taken at different times of irradiation during the reduction reaction of hexacyanoferrate (III) by sodium thiosulfate under irradiation with a Xe lamp equipped with a cutoff filter at 450 nm: (**a**) before irradiation; and after (**b**) 30 min; (**c**) 120 min; and (**d**) 180 min of irradiation.

**Figure 7 materials-12-03012-f007:**
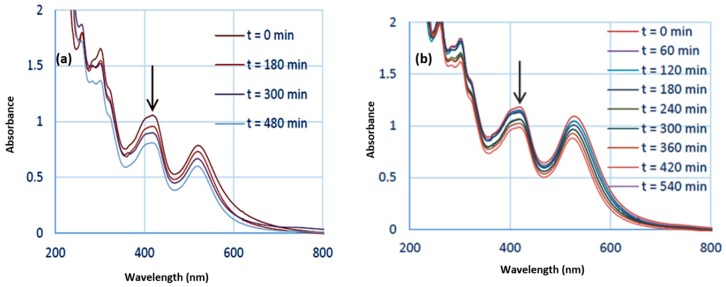
Temporal evolution of the UV-visible spectra of deaerated aqueous solutions containing 5 × 10^−4^ M potassium hexacyanoferrate under LEDs irradiated at 520 nm (**a**) in the presence of 15 nm Au-NPs@citrate and (**b**) 30 nm Au-NPs@citrate. Optical path length: 1 cm.

**Figure 8 materials-12-03012-f008:**
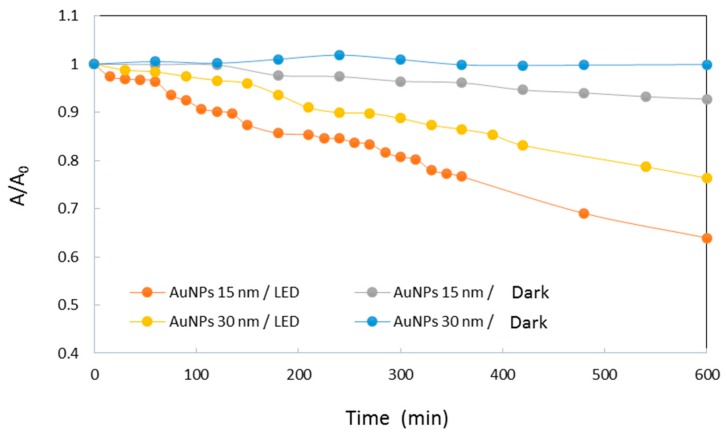
Kinetic traces of the normalized absorbance at 420 nm of HC-FeIII in deaerated aqueous solutions containing 5 × 10^−4^ M potassium hexacyanoferrate in the presence of 15 nm or 30 nm Au-NPs@citrate and in the dark or under LEDs irradiation at 520 nm.

**Figure 9 materials-12-03012-f009:**
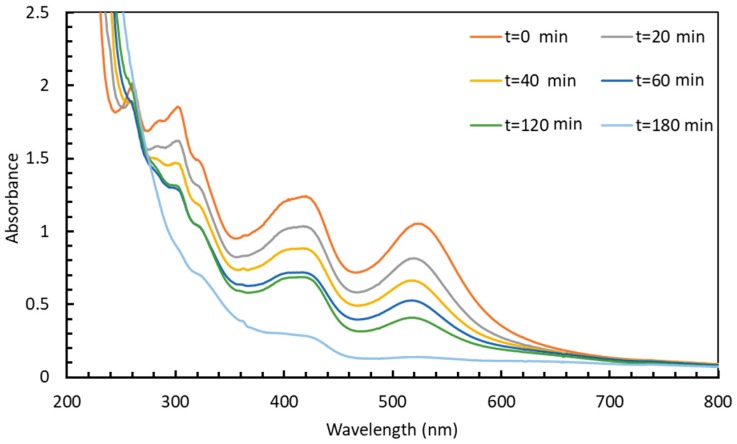
Temporal evolution of the UV-visible spectra of a deaerated aqueous solution containing 5 × 10^−4^ M potassium hexacyano ferrate in the presence of 15 nm Au-NPs@citrate under Xe lamp irradiation (equipped with an optical cutoff filter at 450 nm). Optical path length: 1 cm.

**Figure 10 materials-12-03012-f010:**
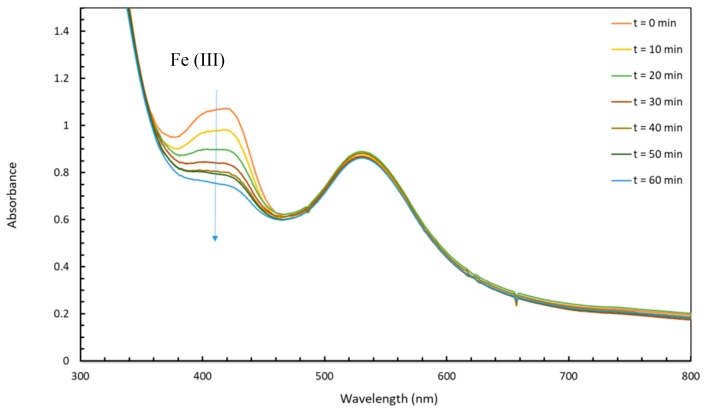
Temporal evolution of the UV-visible spectra of deaerated aqueous solutions containing 5 × 10^−4^ M potassium hexacyanoferrate in the presence of 15 nm Au-NPs@pNTP under LED irradiation at 520 nm (Optical path length: 1 cm).

**Figure 11 materials-12-03012-f011:**
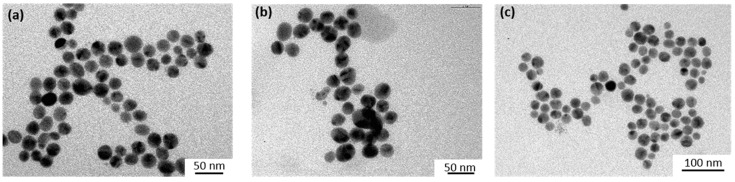
TEM images of the initial 15 nm Au-NPs@pNTP taken at different times during the reduction reaction of hexacyanoferrate (III) under LED irradiation at 520 nm: (**a**) 10 min; (**b**) 30 min; and (**c**) 60 min of irradiation.

**Figure 12 materials-12-03012-f012:**
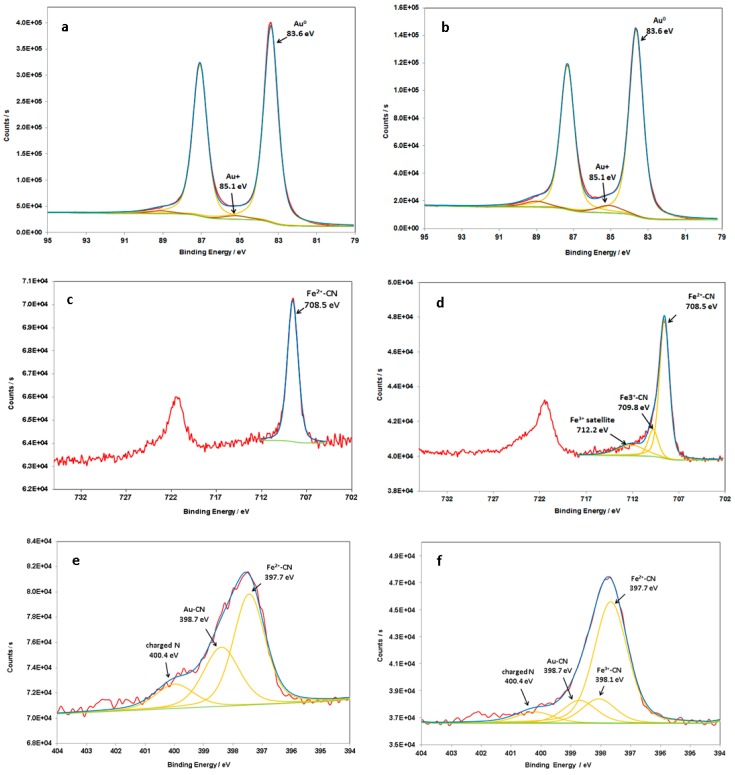
XPS spectra of the Au4f (**a**,**b**), Fe2p (**c**,**d**), and N1s (**e**,**f**) core levels of 30 nm-Au-NPs@citrate after reaction under LEDs irradiation at 520 nm (during 60 min) for deaerated solutions containing hexacyanoferrate (**a**,**c**,**e**) with or (**b**,**d**,**f**) without sodium thiosulfate.

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
