# Peer review of "Hexacyano Ferrate (III) Reduction by Electron Transfer Induced by Plasmonic Catalysis on Gold Nanoparticles"

_materials, 2019, doi:10.3390/ma12183012_

Round 1

Reviewer 1 Report

In this manuscript, the authors describe the reduction of hexacyanoferrate(III) complex by electron transfer induced by irradiated gold nanoparticles. The reaction has been followed by UV-vis spectroscopy and appears clear. The experiments seem clearly described to me and well supported by the experimental data. However, the spectra have not been fully analyzed; for example, the rates of the electron transfer have not been determined and compared, with or without NP on NP irradiation. The kinetic order in the various reagents (or at least with Fe(CN6)) should be determined (is it first order?). Furthermore, as a general comment, I suggest to compare the results obtained in this manuscript with those present in the literature, for similar reactions, like for example those in the presence of borohydrides.

other points:

page 12, line 338. “In Figure 12 c FeII is easily identified by the presence of a single doublet located at 708.5 (Fe2p3/2) and 721.5 eV (Fe2p1/2), typical values for K3[Fe(CN6)] [40].”: but the complex reported as example is a Fe(III) complex.

Figure 12f: why the Fe(III)-CN component is not visible?

Author Response

We thank the referee for his/her very constructive comments.

We analyzed in more details the UV-visible spectra and the kinetic order with respect to Fe(CN)6 has been determined to be first order. Figure 3 has been completed to show the kinetics law. Sentences have been added in p. 4 and 5:

p.4 ” In agreement with Mulvaney’s study [22], the catalyzed reaction is first order with respect to HC-FeIII; as evidenced by the plot of the logarithm of the absorbance of HC-FeIII versus time (Figure 3, bottom); the deduced values of the rate constant are 0.0534 and 0.0244 min-1 for 15 nm and 30 nm AuNPs@citrate, respectively.”

p.5 “Under irradiation the reaction kinetics follows the same order as without irradiation i.e. an apparent first order law with respect to HC-FeIII; the rate constant values are 0.1041 and 0.0679 min-1 for 15 nm and 30 nm AuNPs@citrate, respectively.”

Page 12, line 338, now line 356, there was a typo mistake the compound is K4[Fe(CN)6].

About Fe(III)-CN component, in the presence of ST, at the end of the reaction (after LEDs irradiation for 60 min), there is a complete reduction of FeIII into FeII in the present experimental conditions (as indicated lines 353-354).

We have added a component corresponding to Fe(III)-CN in Figure 12f and modified the text accordingly p. 13: “In the case of theAu-NPs@citrate without sodium thiosulfate (Figure 12f) an additional component corresponding to FeIII – CN is considered. The separation between the contributions FeII-CN and FeIII-CN to N1s is considered to be 0.4 eV [https://srdata.nist.gov/xps], with a ratio in agreement with that of FeIII/FeII obtained from Fe2p spectrum.” Our initial intention was to show only the contributions coming from Fe-CN and Au-CN for clarity but forgot to remove the charge on Fe.

Reviewer 2 Report

This manuscript describes Au nanoparticles (NP)-catalyzed ferricyanide reduction in the presence/absence of thiosulfate. This reaction was found by the other research group in 2009 (ref. 22 in the manuscript) and the reaction itself seems hardly applicable to synthesis and so on. It is almost only new founding that the form of Au-NP changed after high-power irradiation (with Xe-lamp). There are several problematic points in this manuscript as described below.

Photo-irradiation enhanced the reduction in the presence of Au-NP. However, the enhancement is only c.a. double. (see Figure 3). The reaction temperature is missing in the manuscript and supporting information and it is also doubtful whether the temperature was controlled during irradiation. The rising of 20-degree temperature leads the enhancement of the reaction by double. These reactions should be strictly temperature-controlled and the temperature should be added in the manuscript. The author assumes that hot electrons are involved in the case of ferricyanide reduction with Au-NPs@p-nitrothiophenol in the absence of sodium thiosulfate as reductant. However, the solution contained 4x10-4 M p-nitrothiophenol. p-Nitrothiophenol can be much better two or more electrons source than sodium thiosulfate. It is highly possible that p-nitrothiophenol is the electron donor. Bis(p-nitrophenyl)disulfide would be formed. Citric acid can also be good reductant. If hot electrons are involved, the reactivity of the formed electronic hole should be simultaneously considered (that may oxidize ferrocyanide as a back-reaction). A number of typos were found. The author should re-check carefully.

Author Response

We thank the referee for his/her constructive comments that helped us in improving the manuscript.

We have added in the experimental part the temperature at which the experiments were conducted: page 3, lines 115-117: “The reactions were conducted at room temperature (22°C). A thermocouple was used to monitor the temperature during the reaction. Due to absorption of light by all the Au-NPs in the solution, the temperature of the latter raised up to 26°C maximum.”

We have also added two paragraphs p.6 and p.11.

p.6 “As the irradiation was continuous, and considering NP size, light power and irradiation duration, the thermal gradient in the vicinity of each NP is negligible so that local overheating effect can be disregarded [33]. Consequently, the increase of temperature due to plasmon excitation is not enough to favor the reduction power of citrate .”

p.11 “Citrate and para-nitrothiophenol are weak reducing agents. In dark, the reaction does not proceed without thiosulfate for both AuNPs@citrate and AuNPs@pNTP. This indicates that the reduction is not due to para-nitrothiophenol or to citrate. Furthermore the increase in temperature due to plasmon excitation is not enough to favor the reduction power of citrate or that of para-nitrothiophenol [33].”

For the experiments conducted using AuNPs functionalized with pNTP (AuNPs @pNTP), the solutions did not contain much free pNTP because the NPs were centrifuged before catalytic reactions (as now indicated in the experimental part p.3, lines 105-106).

Finally, we hope to have corrected all the typos.

Round 2

Reviewer 1 Report

The authors have improved their manuscript, however I have some concerns regarding the catalytic reactions. They added a sentence (line 115-117) in which they specified that the temperature was not controlled during the tests and an increase of 4 °C is observed. This obviously influences the kinetic of the reaction, so that I suggest to consider only the initial rates of the reaction and not the full reaction time. This is particularly evident in Figure 3 bottom: the data in the first 20-30 minutes can be interpolated with a line different from that using all data (increase in the slope of the line corresponding to an increase of the reaction rate as expected by the increase in the temperature).

If they change this part, the manuscript can be accepted for publication.

Author Response

We thank the referee for his/her very constructive comments.

We removed the sentence “The rate constant values are 0.1041 and 0.0679 min-1 for 15 nm and 30 nm AuNPs@citrate, respectively. and we replaced it by the paragraph p.5-6: “Taken into account only the data in the first 20-30 minutes, the initial rate constant values are determined to be 0.0669 and 0.0475 min-1 for 15 nm and 30 nm AuNPs@citrate, respectively. Indeed, the small increase in temperature measured during the reaction time (£ 4°C) might result in a slight increase in the reaction rate during the reaction time leading to an overestimate of the global rate constant, as suggested by the linear fit in figure 3 (bottom). Moreover, it is to note that the small changes in the LSPR band of the AuNPs@citrate observed during the reaction (Figure 3 (top)) also introduce uncertainties on the determination of the absorbencies due to HC-FeIII, what may affect the values of the rate constant”.

Reviewer 2 Report

There is still unsolved point about temperature control of the reactions.

It is unfortunately revealed that these reactions were not carried out under temperature-controlled condition, because the author described 4 degree’s temperature rising was observed under irradiation in the presence of Au-NP. Water used as solvent is relatively viscous liquid. That is to say, the parts near the Au-NPs should reach much higher temperature than room temperature by irradiation, especially if the solutions were not stirred and enhancement of the reaction rate is apparently due to temperature rising. The author should add the description about whether stirring was carried out in each reaction or not. If not, experiments for kinetics should be re-examined under temperature-controlled condition with stirring.

Author Response

We thank the referee for his/her constructive comments that helped us in improving the manuscript.

The maximum enhancement of the temperature after long irradiation was 3-4 °C. In the revised manuscript we have highlighted (highlight in green) in different paragraphs that all the experiments were conducted under stirring. P.6 “As the irradiation was continuous, and considering NP size, light power, stirring and irradiation duration, the thermal gradient in the vicinity of each NP is negligible so that local overheating effect can be disregarded [33].” Of course, when we put the lighting "on", it heats first in the NPs, but this transient regime lasts a short time compared to the irradiation times. In addition, the density of NPs and the homogeneity of their distribution reduces the gradient. 
